# Detecting Gait Events from Accelerations Using Reservoir Computing

**DOI:** 10.3390/s22197180

**Published:** 2022-09-21

**Authors:** Laurent Chiasson-Poirier, Hananeh Younesian, Katia Turcot, Julien Sylvestre

**Affiliations:** 1Interdisciplinary Institute for Technological Innovation (3IT), Department of Mechanical Engineering, Université de Sherbrooke, Sherbrooke, QC J1K 2R1, Canada; 2Centre Interdisciplinaire de Recherche en Réadaptation et Intégration Sociale (Cirris), Department of Kinesiology, Université Laval, Quebec, QC G1M 2S8, Canada

**Keywords:** gait event detection, reservoir computing, echo state network, IMU sensors

## Abstract

Segmenting the gait cycle into multiple phases using gait event detection (GED) is a well-researched subject with many accurate algorithms. However, the algorithms that are able to perform accurate and robust GED for real-life environments and physical diseases tend to be too complex for their implementation on simple hardware systems limited in computing power and memory, such as those used in wearable devices. This study focuses on a numerical implementation of a reservoir computing (RC) algorithm called the echo state network (ESN) that is based on simple computational steps that are easy to implement on portable hardware systems for real-time detection. RC is a neural network method that is widely used for signal processing applications and uses a fast-training method based on a ridge regression adapted to the large quantity and variety of IMU data needed to use RC in various real-life environment GED. In this study, an ESN was used to perform offline GED with gait data from IMU and ground force sensors retrieved from three databases for a total of 28 healthy adults and 15 walking conditions. Our main finding is that despite its low complexity, ESN is robust for GED, with performance comparable to other state-of-the-art algorithms. Our results show the ESN is robust enough to obtain good detection results in all conditions if the algorithm is trained with variable data that match those conditions. The distribution of the mean absolute errors (MAE) between the detection times from the ESN and the force sensors were between 40 and 120 ms for 6 defined gait events (95th percentile). We compared our ESN with four different state-of-the-art algorithms from the literature. The ESN obtained a MAE not more than 10 ms above three other reference algorithms for normal walking indoor and outdoor conditions and yielded the 2nd lowest MAE and the 2nd highest true positive rate and specificity when applied to outdoor walking and running conditions. Our work opens the door to using the ESN as a GED for applications in wearable sensors for long-term patient monitoring.

## 1. Introduction

The functional analysis of human locomotion allows the gait cycle to be defined in time intervals [1] called gait phases. They are distinguished by specific gait events. Gait event detection (GED) can thus be considered a fundamental processing step needed for gait analysis. The two main gait events are the heel strike (HS) and the toe off (TO), which, respectively, define the beginning of the stance phase and of the swing phase. Multiple clinical analysis and applications use gait events, such as foot clearance estimation [2], to monitor the physical behaviour of participants with neurological [3,4] or osteoarthritic [4] disease. Some of the clinical applications requiring real-time GED include functional electrical stimulation [5,6] and gait retraining [7,8]. Among several technologies used for GED, user-friendly, light, wireless and compact inertial measurement unit (IMU) sensors are crucial to provide follow-up in the daily movements of the participants. IMU sensors have thus become an essential tool for research and diagnosis. However, the accelerations captured by an IMU sensor are exposed to variable movements (nominal cadence, regularity, symmetry) inherent to the ecological environment and pathological walking conditions. Pathologies, disease progression and participant fatigue contribute to the movement variation and the speed of the gait. External conditions, such as the type of ground and the walking trajectory, can also change the gait dynamics. Several complex IMU-based GED processing algorithms have therefore been widely explored over the past decade to increase the precision and the robustness of GED under various walking conditions. GED algorithms use different strategies, such as biomechanical models [9], empirical filtering and signal processing methods [5,10,11,12,13,14], time-frequency analysis [15,16], symbol- or statistic-based methods [17,18] and hidden Markov models [19]. Most algorithms need to be designed for acquisition protocols that use only one sensor position on the body to reduce complexity and cost [5,10,12,15,19,20].

The problem with the methods reported thus far is that wearable systems continuously tend to become smaller and more autonomous to allow good performance in long-term ecological settings. To do so, algorithms stored in the device have to be light in memory and in calculation. Moreover, ecological conditions that increase the variability in the data also increase the processing complexity of most current algorithms. For example, biomechanical models or empirical filtering methods have limited usability in ecological conditions because they often require parameter changes when new gait conditions are considered [5,21]. On the other hand, a robust GED algorithm such as [15] showed very high accuracy for many different gait conditions, but uses a time-frequency transform and a two-dimensional Gaussian curve fit in the time-frequency plane to identify HS and TO events, which can be difficult to implement in small portable devices.

Moreover, previous studies are limited by most algorithms being designed to only distinguish between two gait phases by detecting two gait events: the HS and the TO [10,12,13,15,16]. They distinguish the phase when the foot is in contact with the ground (stance phase) from when it moves (swing phase). Some applications need to detect a higher number of gait events inside the stance and the swing phases, as is the case for functional electric stimulation (FES), which used for patients with neurological diseases such as Parkinson’s or in post-stroke readaptation [5,6,22]. Additionally, the inner phase of the stance can be a meaningful temporal parameter to identify differences between normal and pathological gaits by observing patients before and after surgical treatments for ankle osteoarthritis [23].

The use of artificial neural networks (ANNs) in gait analysis is currently gaining popularity, especially for human activity recognition (HAR) tasks [24,25,26,27], which are increasingly important in human-machine interactions. ANN show good performance when dealing with the large amount of variability from the environment (hardware acquisition system, the walking surface, noise) and the subject (clothing, shoes, carrying items, walking speed, direction, age, gender or physiological conditions) [24]. Some studies that address the use of ANNs for HAR tasks also used GED for gait cycle normalization in order to increase the performance of the recognition [24]. In addition, affordable IMU sensors and force-sensitive resistors (FSR) enable an easier large-scale gait data collection to apply ANN directly for GED [28].

This paper presents a new approach for GED by using a class of ANN algorithms called reservoir computing (RC) [29]. Recently, ANN algorithms have become prominent due to highly accessible digital data [30]. RC specializes in processing temporal signals such as accelerations. It can also be implemented quickly with a fast and systematic training methodology to adjust coefficients using a linear regression. This means that it can be adapted quickly for different ecological conditions. No other algorithms seem to be as suitable to meet the requirements of low computational complexity necessary for hardware implementation as well as robustness over a wide range of ecological conditions.

In this study, we use the numerical implementation of RC called the echo states network (ESN) [31]. Many applications use the ESN algorithm [32], but only a few are applied to biomechanics, such as gesture recognition [33], muscle drive-in actuation [34] and exoskeleton control [35]. The ESN model is much simpler to implement on hardware devices compared to most of the advanced GED models recently developed. It can be applied to many walking conditions without increasing the algorithm complexity. To the best of our knowledge, the current literature does not address the application of ESN for GED.

As other ANN algorithms, RC models can be seen as black box models that learn how to process an input to give a useful output prediction. The RC model is applied using a supervised training method, where the predictions are adjusted to be as close as possible to a useful target that is hard to compute otherwise. In this study, we exploited the RC to learn the best relationship between example datasets of input and output target pairs, with the inputs being the acceleration signals and the output being binary indicator signals indicating the gait events. This method is separated in two steps: training and testing. The training step adjusts the trainable parameters of the RC model to establish the optimal relationship between the targets and the RC model predictions. We used many examples of input-target pairs based on experimental measurements. The parameters of the RC model are adjusted to minimize the error between the prediction computed by the RC and the target. The testing step evaluates the capability of the RC to correctly predict the target from new and unseen data (testing dataset). This indicates how well the RC will perform in operation. We measured the RC performance by the error between the model predictions and the targets of this new independent dataset.

The novelty of our study is that it demonstrates the benefits of the ESN approach for GED by adjusting the model for the detection of multiple events (in addition to HS and TO) over multiple walking conditions using a fast training methodology. No GED algorithm in the literature had the ability to offer similar adaptability with the low complexity needed for hardware implementation in simple wearable devices. Our objective is to characterize the ESN performance for GED for six types of event in the gait cycle. We considered different aspects impacting its performance: the characteristics of the training data used, the nature of the sensor data (orientations and positions on the body and the ground walking conditions), as well as the detection capability of the different types of events. Multiple training and testing datasets from three different databases are used to understand the impact of each of these aspects. We also compared the performance of the ESN to other algorithms in the literature detecting the HS and the TO gait events.

The methodology presents how to collect, format and process the gait data using a reservoir computer algorithm. Section 2.1 introduces the three databases of experimental gait data used for our experiments. Section 2.2 explains the process to prepare the gait data in a usable format for the ESN training and testing. Section 2.3 presents the ESN algorithm in detail, and Section 2.4 presents a comparative GED algorithm tested in this work. Section 2.5 presents the standard metrics used to evaluate the GED algorithm performance. Section 3 presents the results of the ESN and the comparative algorithms. Finally, Section 4 is a discussion about the advantages and limitations of the ESN, and it establishes a comparison of the ESN performance with the other algorithms from the literature.

## 2. Methods

The evaluation process of the GED performance of the ESN and the Teager–Kaiser energy operator (TKEO) algorithms is presented in the flowchart of Figure 1. In this process, we used experimental data of the acceleration and ground reaction force measured on participants during variable walking tests, as detailed in Section 2.1. We applied preprocessing steps, including identifying gait events from force sensors signals, structuring the data in timeseries and partitioning the data in training set and a testing set, as explained in Section 2.2. We used a supervised training methodology, which implies that the evaluation process of the ESN was separated into two main processing steps: training and testing. As explained in Section 2.3, the training step established the mean gait cycle duration of the gait data (τ) and the output weight matrix ($W_{\mathrm{out}}$) of the ESN. The testing step of the ESN predicted the gait events using the peaks in its output signal (**Y**). We used the TKEO GED algorithm [10], presented in Section 2.4, for comparison purpose in the testing step. We established the ESN and the TKEO’s performance with the error between the prediction events and the target events, expressed as the positive prediction value (PPV), the true positive rate (TPR), the T1 score and the mean absolute error (MAE) between event times, as explained in Section 2.5.

### 2.1. Data Collection

We use three databases to test the ESN under a large variability of the walking conditions, namely the MAREA, CIRRIS and UDS database, all including gait acceleration and ground force signal timeseries recorded on healthy adults. The MAREA gait database (MAREA: Movement Analysis in Real-world Environments using Accelerometers https://wiki.hh.se/caisr/index.php/Gait_database, (accessed on 4 February 2021)) is an open-source database of accelerations and prelabelled gait events recorded under various walking and running conditions that has been already used to test various GED algorithms [10,15,20]. We also assembled the CIRRIS and the UDS databases for the purpose of this study, respectively, at the CIRRIS research center and at the Université de Sherbrooke. All participants in the CIRRIS and UDS databases gave written informed consent to participate in the study according to experimental procedures approved by the IRDPQ-CIUSSS-CP research ethical committee (2021–2269) (IRDPQ-CIUSSS-CP: Institut de Réadaptation en Déficience Physique de Québec et Centre Intégré Universitaire de Santé et de Services Sociaux de la Capital Nationale). Because of the different acquisition frequencies in each database, the data were resampled at 200 Hz using polyphase filtering [36]. Each database was established under independent conditions (treadmill, indoor, outdoor, pathology simulation, obstacles). Table 1 gives an overview of all conditions recorded in each database, including ground conditions, number of participants, duration and walking speed. Details of the data collection process for each database are given below.

#### 2.1.1. MAREA Database

The MAREA database was collected on 11 subjects under 5 indoor conditions: flat treadmill walking (TW) and running (TR), inclined treadmill (TI), indoor flat walking (IW) and running (IR). The database also covers 9 subjects under 2 outdoor conditions: outdoor walking (OW) and running (OR). Each subject had a 3-axis Shimmer3 accelerometer (±8 g) attached to their waist, left wrist and left and right ankles (Figure 2). Two force-sensitive resistors were also placed under the foot, at the heel and the toes, to acquire foot pressure measurements on the ground.

#### 2.1.2. CIRRIS Database

The CIRRIS database was recorded on two participants during normal walking (N) and while subjects were simulating 3 different gait strategies related to knee osteoarthritis [37,38,39]: right toe with an external angle (TA), flat strike of the right foot (rFF) and right trunk leaning (rTL). For each randomized condition, both participants walked for 2 min at 5 different speeds (0.8, 1, 1.2, 1.4, 1.6 m/s). A 3-axis Physilog 4 accelerometer (±8 g) was placed on the top of both shoes, localized as shown in Figure 2. Ground reaction forces for each foot were recorded separately using a treadmill with two embedded force plates (Bertec, Columbus, OH, USA, 1000 Hz). All signals were synchronized using a heel impact on the treadmill.

#### 2.1.3. UDS Database

The UDS database was recorded on 15 participants in three walking circuits in an indoor gym. The circuits were a 12 m traverse on a gym track. Three different tracks were used in the tests, as shown in Figure 3: the normal (T1), the obstacle (T2) and the walk around (T3) tracks. Each participant was equipped with an Adafruit MMA8451 Triple-Axis Accelerometer (±2 g) placed on the top of the left foot inside the shoe (Figure 2). Force-resistive sensors were placed under the left shoe, under the shoe sole. Records of 2 min of continuous round trip walking without stopping at the end were done for each track and each participant. The data of T3 for one participant was not taken into account due to incorrect positioning of the FSR.

### 2.2. Data Formatting and Preprocessing

#### 2.2.1. Target Gait Events Identification

The target gait events (GE) were defined by time indices of different peaks or drops of the foot ground reaction force. These indices were identified with FSR measurements for the MAREA and UDS databases and intrumented treadmill measurements for the CIRRIS database. Six GE classes, the heel strike, (HS), heel push (HP), foot flat (FF), heel off (HO), toe push (TP) and toe off (TO) were defined in the gait cycle, as shown in Figure 4. All GE classes were located in the stance phase, and one GE index was defined in each gait cycle.

Only some of the six gait event classes were identified in each database. For the MAREA database, only HS and TO were identified directly by the authors of the data source [15]. The HS, HP, FF, TP and TO event classes were determined for the CIRRIS database, and the HS, HO, TP and TO event classes were determined for the UDS database. Processing details of the ground reaction force signal to obtain the targets gait event are given in Appendix C.

#### 2.2.2. Input and Target Timeseries Definition

The acceleration and gait event targets were recorded separately for each subject and each walking conditions. Different data subsets grouping all the records for one or more types of walking conditions were formed as presented in Table 1. All records in each subset were identified by the index k. For example, in the subset ${\mathrm{MAR}}_{\mathrm{all}}$, *k* = 1 corresponds to the record of the first participant tested in the treadmill walking condition, *k* = 2 corresponds to the data from the second participant in the same condition, and *k* = 12 corresponds again to the first participant, but now tested in the treadmill running condition. $U_{k}$ corresponds to the acceleration signal of length T${}_{k}$ (with T${}_{k}$ the number of time points in the recording). $U_{k}$ may have only one row corresponding to one of the accelerometers. When various axes are considered, $U_{k}$ has multiple rows, each corresponding to an individual axis.

$Y_{k}^{\prime }$ corresponds to the gait event targets signal of length T${}_{k}$ and is a binary signal, as shown in Figure 1. A value of 1 is inserted in $Y_{k}^{\prime }$ at the time index of each target gait event identified from ground force mesurements, and otherwise, $Y_{k}^{\prime }$ is set to 0. $Y_{k}^{\prime }$ has multiple rows, each corresponding to one of the GE classes (HS, HP, FF, HO, TP, TO). For example, in ${\mathrm{MAR}}_{\mathrm{all}}$, since only the HS and the TO were detected, $Y_{k}^{\prime }$ had two rows. The first row had a value of 1 only at each HS event index identified, and the second row had a value of 1 only at each TO event index identified; other entries were 0.

#### 2.2.3. Subset Partitioning of the Timeseries

The training and testing process in Figure 1 was performed iteratively with multiple subset selections of walking conditions, as presented in the rightmost column of Table 1. These subsets defined the two lists of input timeseries $\left\{U_{1},...,U_{k},...,U_{N_{\mathrm{TS}}}\right\}$ and target timeseries $\left\{Y_{1}^{\prime },...,Y_{k}^{\prime },...,Y_{N_{\mathrm{TS}}}^{\prime }\right\}$. $N_{\mathrm{TS}}$ is the total number of timeseries and varies for each subset selection. The results presented in Section 3.1, Section 3.3 and Section 3.5.1 are based on the subsets ${\mathrm{MAR}}_{\mathrm{all}}$, ${\mathrm{CIR}}_{\mathrm{all}}$ and ${\mathrm{UDS}}_{\mathrm{all}}$, where all timeseries of each database were used. In Section 3.2, we use subsets selecting a few specific walking conditions to evaluate the influence of the walking condition variability on the performance of the ESN algorithm. We also used specific condition subsets in Section 3.5.2 to compare the ESN with the TKEO and three other algorithms from the literature.

#### 2.2.4. Training and Testing Sets Partioning

We partitioned the timeseries of accelerations and gait event targets in two sets by taking 70% of the total number of gait cycles for the training set, while the other was assigned to the testing set. We defined the separation index in the middle of the stance phase (MSP) using the gait event classes in each database. For the MAREA database, the MSP was defined by the means indices between HS and TO. For the UDS and CIRRIS databases, the MSP was defined with the HO and the FF, respectively. We selected the separation index at the closest MSP to 70% of the length of the whole timeseries. We eliminated the data before the first MSP in training signal to ensure that both training and testing signals started at the same place in the gait cycle (the MSP). This preprocessing ensures the removal of a uniform gait cycle in each timeseries, considered as transient data in the ESN method, as explained in Section 2.3.1.

### 2.3. ESN Procedure

#### 2.3.1. Main ESN Structure

We present here the ESN algorithm presented by Lukoševičius [31] applied for GED using recorded body accelerations. The ESN is a neural network build from three layers: the input layer, the hidden layer and the output layer, as shown in Figure 5. The hidden layer is commonly called the reservoir and is composed of N artificial neurons. The ESN processes the input accelerations u(n) and returns the predictions y(n) for the GE according to
(1)$$\tilde{x}\left(n\right)=\tanh\left(W_{\mathrm{in}}u\left(n\right)+Wx\left(n-1\right)\right)$$
(2)$$x\left(n\right)=\alpha \tilde{x}\left(n\right)+\left(\alpha -1\right)x\left(n-1\right)$$
(3)$$y\left(n\right)=W_{\mathrm{out}}x\left(n\right),$$
where u(n) is a column vector of size $N_{\mathrm{in}}$, and where $N_{\mathrm{in}}$ corresponds to the number of acceleration channels in the input augmented by 1 to add a bias value. *n* is the time-step index. y(n) is a column vector of size $N_{\mathrm{out}}$. $W_{\mathrm{out}}$ is the output matrix of size $N_{\mathrm{out}}$×N, where N is the number of neurons in the reservoir, and $N_{\mathrm{out}}$ is the number of gait event classes predicted. When the model is used only for HS and TO events, $N_{\mathrm{out}}$ = 2. $N_{\mathrm{out}}$ = 2, 4 and 5, respectively, for the events for the MAREA, UDS and CIRRIS databases when all events are used. x(n) is a column vector of size N corresponding to the states of the individual artificial neurons in the reservoir. The variable α is the leaking rate of the reservoir. It can be set between 0 and 1 and is used to adjust the response time of the ESN. The node connection matrix W of size N × N and the input layer matrix $W_{\mathrm{in}}$ of size N×$N_{\mathrm{in}}$ are fixed and are initialized from random uniform distributions. A distribution between [−1, 1] is used to set the elements of the matrices W and $W_{\mathrm{in}}$. The sparsity of W is adjusted by setting to zero a fraction 1-P of randomly chosen elements, where P is the sparsity. The spectral radius, ρ, is set by dividing all elements of W by its absolute maximum eigenvalue and multiplying it by ρ. For the input layer matrix $W_{\mathrm{in}}$, its sparsity is adjusted by setting a fraction of 1-$P_{i}$ of randomly chosen elements to zero. The input bias scaling of the matrix $W_{\mathrm{in}}$ is adjusted by multiplying its first row by $S_{b}$. The input scaling is adjusted by multiplying all rows but the first of $W_{\mathrm{in}}$ by $S_{i}$. The parameters used to set the ESN are shown in Table 2. We set these parameters to be fixed for all results. They were chosen based on an exploration of the CHARC properties [40], as presented in Appendix A.

Equations (Equation 1)–(Equation 3) were iterated for n running over all the time-steps of each input timeseries $U_{k}=\left[u\left(1\right),\;u\left(2\right),\;...\;u\left(T_{k}\right)\right]$ of size $N_{\mathrm{in}}$×$T_{k}$, where T${}_{k}$ is the total number of time-steps for the input timeseries $U_{k}$. We forced the states to zero at the first time-step, x(0)=0. For the indices of the first gait cycle, the reservoir states x(n) were different compared to other gait cycles since the initial value was 0. We considered the time indices of the first 1.25 s (n≤250) as a transient phase, and we eliminated them before further processing. We concatenated the states x(n) of the reservoir over all the time-steps of one timeseries to form the states matrix $\chi_{k}=\left[x\left(251\right),\;x\left(252\right),\;...\;x\left(T_{k}\right)\right]$ of size N × (T${}_{k}$-250).

#### 2.3.2. ESN Training

The training of the ESN fixes the weights of the output matrix $W_{\mathrm{out}}$ and determines the gait cycle duration τ, defined as the mean duration between each HS gait events. As explained in Section 2.2.3, the ESN processes input accelerations from multiple timeseries, $\left\{U_{1},\ldots ,U_{k},\ldots ,U_{N_{\mathrm{TS}}}\right\}$, each of size $N_{\mathrm{in}}$×$T_{k}$. Equations (1) and (2) are first used to compute the reservoir state matrices $\chi_{k}$ for each input timeseries of the training dataset.

We defined the same number of target matrices (output signal excluding the first 1.25 s), $\left\{Y_{1}^{\prime },...,Y_{k}^{\prime },...,Y_{N_{\mathrm{TS}}}^{\prime }\right\}$, each of size $N_{\mathrm{out}}$ × (T${}_{k}$-250). We then computed the $XX^{T}$ and $Y^{\prime }X^{T}$ using the summation of premultiplied state matrices for each individual timeseries as $XX^{T}=\sum_{k=1}^{N_{\mathrm{TS}}}\chi_{k}\chi_{k}{}^{T}$ and $Y^{\prime }X^{T}=\sum_{k=1}^{N_{\mathrm{TS}}}Y_{k}^{\prime }\chi_{k}{}^{T}$, where $\chi_{k}{}^{T}$ is the transposition of the matrix $\chi_{k}$. (The matrix **X** of size **N** × T${}_{\mathrm{tot}}$, is the matrix of all $\chi_{k}$ concatenated together, and **Y**, of size $N_{\mathrm{out}}\times$$T_{\mathrm{tot}}$, is the matrix of all $Y_{k}^{\prime }$ concatenated together, where T${}_{\mathrm{tot}}$ is the total number of time-steps over all the timeseries, $T_{\mathrm{tot}}=\sum_{k=1}^{N_{TS}}T_{k}$. In practice, we are not computing these matrices **X** and **Y** explicitly, because if all state matrices $\chi_{k}$ and $Y_{k}^{\prime }$ are kept in memory in order to build the matrix **X**, this quickly induces memory saturation since many timeseries are considered.) $W_{\mathrm{out}}$ is computed using a ridge regression,
(4)$$W_{\mathrm{out}}=Y^{\prime }X^{T}{\left(XX^{T}+\gamma I\right)}^{-1},$$
to minimize the mean square error between the output target events and the predicted event timeseries. The matrix I (N × N) is the identity matrix, and γ is a regularization parameter used to prevent overfitting. The parameter γ in Equation (Equation 4) was adjusted to $1\times 10^{-6}$ (Table 2) by trying multiple values on a logarithmic scale and keeping the one giving the best NRMSE according to Equation (5) using the ${\mathrm{MAR}}_{\mathrm{all}}$, ${\mathrm{CIR}}_{\mathrm{all}}$ and ${\mathrm{UDS}}_{\mathrm{all}}$ as a validation set.

Finally, the mean gait cycle duration, τ, was established from the HS gait event target of all the timeseries in the training set. A lower range gait cycle duration for each timeseries, $\tau_{k}$, was computed using the 5th percentile of the time distances separating each HS events indicated in $Y_{k}^{\prime }$. The value τ is computed as the mean of each timeseries gait cycle duration, $\tau =\frac{1}{N_{\mathrm{TS}}}\sum_{k=1}^{N_{\mathrm{TS}}}\tau_{k}$. The value τ was equal to 0.96, 1.17 and 1.19 s per step, respectively, for the MAREA, CIRRIS and UDS databases considering subsets with all the conditions selected (${\mathrm{MAR}}_{\mathrm{all}}$, ${\mathrm{CIR}}_{\mathrm{all}}$ and ${\mathrm{UDS}}_{\mathrm{all}}$). The MAREA database includes more running conditions than CIRRIS and UDS, which explains the lower value of τ.

#### 2.3.3. ESN Testing

During the testing, we used the Equations (1)–(3) with the $W_{\mathrm{out}}$ learned in the training phase to compute a prediction output $Y_{k}$ from the ESN for each input timeseries of the testing dataset. $Y_{k}$ was a multichannel matrix of continuous signals with the same size as the gait event target matrix $Y_{k}^{\prime }$. Each channel in $Y_{k}$ was the prediction for one gait event class. We evaluated the error of the ESN by the normalized root mean squared error (NRMSE), defined as
(5a)$${\mathrm{NRMSE}}_{k}=\sum_{i=1}^{N_{\mathrm{out}}}\sqrt{\frac{1}{T}\frac{\sum_{n=1}^{T}{\left(Y_{k,i,n}-Y_{k,i,n}^{\prime }\right)}^{2}}{\sigma_{k,i}^{2}}},$$
(5b)$$\mathrm{NRMSE}=\frac{1}{N_{\mathrm{TS}}}\sum_{k=1}^{N_{\mathrm{TS}}}{\mathrm{NRMSE}}_{k},$$
where *k* is the index of the input timeseries, and *i* and *n* are the row and column indices, respectively, related to the event type and the time-step in $Y_{k,i,n}^{\prime }$ and $Y_{k,i,n}$. $\sigma_{k,i}$ is the standard deviation of the *i*${}^{\mathrm{th}}$ column target signal $Y_{k,i}^{\prime }$.

We finally used a peak finder algorithm [41] to obtain the time indices of the peaks of each prediction channel signal, $Y_{k,i}$. The time indices of the peaks in each channel corresponded to the predicted time of the gait event according to each gait event class. We used a threshold of 0.65τ to limit the minimum allowed time between two peaks from a given event class in order to minimize the misdetection of events, as explained in Appendix B.

### 2.4. Comparative Procedure

One of the objectives of this work was to establish a performance comparison between an ESN and other IMU GED algorithms. The Teager–Kaiser energy operator (TKEO) method proposed by Flood et al. [10] uses an energy operator to determine the two energy peaks associated with the initial contact and the final contact of the foot on the ground, referred here as the HS and the TO gait events. This method has demonstrated good performance on the MAREA walking database for the mean absolute error metric (MAE) [10].

Similar to the ESN, the TKEO method has the advantage of minimizing the number of assumptions about gait (e.g., foot symmetry, regularity) to process events. In this paper, the TKEO method was implemented following the indications of [10].

The TKEO method consists of a two-step nonlinear filtering process executed individually on each timeseries $U_{k}$ of the test set. The first step computes a timeseries $\Phi_{k}$ of size T${}_{k}$ representing the amount of energy at each time point of the acceleration $U_{k}$, expressed in terms of the function $F_{TKEO1}$ [10],
(6)$$\Phi_{k}=F_{TKEO1}\left(U_{k}\right),$$
where $U_{k}$ is a single channel of acceleration. The HS events are detected from the indices of the peaks in $\Phi_{k}$ using a peak finder algorithm [41] with a time detection threshold defined as 0.7 of the first autocorrelation peak of $\Phi_{k}$ [10]. As a second step, a function $F_{TKEO2}$ [10] computes a modified energy function $\Psi_{k}$, also of size T${}_{k}$,
(7)$$\Psi_{k}=F_{TKEO2}\left(\Phi_{k},{\left\{i_{HS}\right\}}_{k}\right),$$
based on the energy $\Phi_{k}$ and the HS event indices ${\left\{i_{HS}\right\}}_{k}$ in order to eliminate the peaks previously detected for the HS in $\Phi_{k}$ and to smooth the signal. The TO events are detected from the indices of peaks in $\Psi_{k}$ using the same peak finder algorithm used for the HS events and with a time detection threshold equal to 0.7 of the first autocorrelation peak of $\Psi_{k}$. We tested our implementation of this algorithm with the same reference datasets as in [10]. As shown in Figure 6, there is a good agreement.

In order to compare the precision of the ESN, TKEO and other models from the literature, we removed the median bias from the errors in all results for all methods. (A median bias in the overall prediction can appear for any type of GED algorithm and can be easily removed in applications without changing the GED algorithm performance.) This reduced the MAE obtained by the TKEO compared to the value reported in the literature [10].

Small variations in the standard deviation were observed compared to the value reported by Flood [10]. This could be related to our implementation in the Python programming language, where the peak finder algorithm [41] was different compared to the one used in the original MATLAB implementation [42]. Since the values of the MAE obtained were mostly the same compared to the reference, our implementation was considered as a valid comparisons algorithm of the TKEO reported by [10] tested here on all our databases.

### 2.5. Performance Criteria

The metrics of GED performance are based on the error between the time positions of the target and the predicted events. For each timeseries *k* and class of event i, the vector of the error values **E**${}_{ik}$ (each of size $1\times N_{TE}\left(i,k\right)$) was calculated by taking the time of each target event, minus the time of the closest prediction event. $N_{TE}\left(i,k\right)$ is the number of target events in the channel of the target vector $Y_{k}^{\prime }$, and $N_{PE}\left(i,k\right)$ is the number of predicted gait events detected from the GED. The two can be different for a gait event class *i* and timeseries *k*.

As proposed in other works on GED [15,20], we removed the constant bias in the error before assessing the performance metrics for each class of events by subtracting the median of the error over all timeseries. Using the absolute values of the vector **E**${}_{ik}$, we classified the target and prediction events into three detection outcomes following the rules presented in Table 3. A threshold of 65 ms was chosen to consider the target events correctly detected, similarly to [10]. For each class of event i, the number of true positives (TP), false positives (FP) and false negatives (FN) were cumulated for all timeseries *k*. We express results with standard metrics of the detection rate used in GED, the true positive rate or sensitivity (TPR),
(8)$${\mathrm{TPR}}_{i}=\frac{N_{i}^{TP}}{N_{i}^{TP}+N_{i}^{FP}}$$
and the positive prediction value or precision (PPV),
(9)$${\mathrm{PPV}}_{i}=\frac{N_{i}^{TP}}{N_{i}^{TP}+N_{i}^{FN}}.$$

These metrics are generally anti-correlated: oversensitive algorithms will have a low TPR but a high PPV, and vice versa for undersensitive algorithms. The $T_{1}$ score represents the harmonic mean of the TPR and PPV,
(10)$$T_{1}=\frac{2\;\mathrm{TPR}\;\mathrm{PPV}}{\mathrm{TPR}+\mathrm{PPV}},$$
where a $T_{1}$ result of 1 represents perfect detection without any FP and FN. Another metric that is used in the literature is called the specificity (*SPF*),
(11)$$SPF=\frac{N_{i}^{TN}}{N_{i}^{TN}+N_{i}^{FP}},$$
which measures the oversensitivity of an algorithm. Similar the TPR, the value of the *SPF* will be low if the algorithm is oversensitive. This metric is used to compare the ESN with other algorithms from the literature in Section 3.5.2. However, the *SPF* is not used to express the results in the other parts of the paper because its value is influenced by the ratio between the number of gait cycles and the total number of points in the timeseries. The walking speed and the sampling frequency will then influence this proportion and the *SPF* result, even if the algorithms have the same range of errors. Comparing different walking conditions or other works in the literature is then less robust.

The GED algorithm’s performance was also measured using the mean absolute error (MAE). For each timeseries *k* and event class *i*, the mean absolute error $\mathrm{MAE}{}_{ik}$, in seconds, is calculated as the mean of the absolute of the error vector $E_{ik}$ (already recentered for each event by the median error over all timeseries). The $\mathrm{MAE}{}_{ik}$ results are grouped by gait event class, and the distributions over all timeseries are displayed in whisker plots.

## 3. Results

We present results in five sections. In Section 3.1, we focus on the influence of the training data (training length, choice of the database in training versus testing) used to evaluate the performance of the ESN. In Section 3.2, we evaluate the influence of the walking conditions on HS and TO events using the antero-posterior (AP) acceleration of the foot. In Section 3.3, we focus on the choice of axis: AP, medio-lateral (ML) and vertical (V), as well as various sensor positions (foot, wirst and waist), as the input signal of the model. In Section 3.4, we compare the performance between all the event classes. Finally, in Section 3.5, we compare the ESN performance with other GED algorithms in the literature on the MAREA database based on the prediction of the HS and TO events.

### 3.1. Training Data

This section evaluates the influence of two aspects of the training data on the ESN performance. The NRMSE and the MAE results are presented as a function of the amount of training data in Section 3.1.1. The MAE (mean and standard deviation) for each dataset is presented for various training and testing sets in Section 3.1.2.

#### 3.1.1. Training Data Length

Figure 7 shows the NRMSE and MAE as a function of the length of the training data. All timeseries in the testing set were kept with a fixed length equal to the last 30% part of the full recorded data, as explained in Section 2.2.4. The MAE is presented with the 50th and 75th percentiles of all the $\mathrm{MAE}{}_{ik}$ of all gait event classes *i* and timeseries *k*. Looking at the 75th percentile of the MAE, the performance of the ESN converges for a total training time of approximately 2000 s. The training for the three databases combined (ALL) did not need more time to converge than each individual database. The NRMSE for the three databases combined is 0.980 ± 0.002 after convergence. Considering that the average cadence in the different databases is about 0.9 steps/s, 2000 gait cycles is a good standard to train the ESN to obtain the best precision for GED. We observed both the NRMSE and the MAE as a convergence criterion to establish the minimum training size, but the MAE metric is more representative of the ESN performance for GED.

In addition, we see in Figure 7 that the NRMSE keeps decreasing with longer training even if performance in terms of MAE stays approximately the same. In many machine learning applications, the NRMSE is a gold standard to evaluate the training convergence. Here, the NRMSE continues to decrease with training time much longer than 2000 cycles before obtaining a stable value. However, the NRMSE expressed the error of all points between the prediction signal and the target binary signal. The positions in time and the sharp shape of the peaks in the prediction signal can influence the NRMSE. Results show that even if the NRMSE decreased with higher training length over 2000 gait cycles of training data, the ESN did not perform better for GED in terms of MAE. The MAE calculation considers only the time position of each detected peak. Even if the noise of the prediction signal might decrease with more training data, the MAE would stay the same if the positions of these peaks stayed the same. Thus, the MAE is a better indicator of the performance than the NRMSE for GED.

#### 3.1.2. Training between Datasets

We show the median and interquartile range of the MAE for different training and testing sets in Figure 8. To allow the use of the ESN with training data timeseries from a different database, the predictions considered only two classes of events identified for all three databases, the HS and TO. All timeseries in each database were considered. The results on the diagonals of the matrices in Figure 8 correspond to the case where the training set and the testing sets were coming from the same database. The off-diagonal results correspond to the cases where the training and the testing sets came from different databases (the first 70% of the timeseries from one database was used for the training set and the last 30% of the timeseries from another database was used for the testing set). The first three rows correspond to the cases where only one of the three databases was used in the training and the last row (or the last column) corresponds to the case where all databases were used for the training (or the testing).

As expected, the best results were obtained when the training and testing data came from the same population. We obtain intermediate results when the training was conducted using all databases. The ESN performance was not quite as good as when the training and testing database matched exactly, but its error remained under a value of MAE of 20 ms. High errors were obtained when the training was carried out using a different database than the one used for testing.

### 3.2. Performance under Various Walking Conditions

Figure 9 shows that the ESN algorithm is robust enough to perform similarly under all of the walking conditions tested in this work. The prediction results for variable walking conditions are shown for the HS and TO gait events, common to the three databases. Each category along the horizontal axis represents the result for a specific subset of walking condition considered in the train and test sets, as presented in Table 1.

These categories showed that the MAE distributions increased slightly when more walking conditions were included in the subsets. In the MAREA database, the prediction performance was mostly equivalent between indoor conditions (${\mathrm{MAR}}_{\mathrm{TW}}$, ${\mathrm{MAR}}_{\mathrm{TR}}$, ${\mathrm{MAR}}_{\mathrm{IW}}$, and ${\mathrm{MAR}}_{G1}$) and outdoor conditions (${\mathrm{MAR}}_{\mathrm{OW}}$ and ${\mathrm{MAR}}_{\mathrm{OR}}$). Mixed conditions (${\mathrm{MAR}}_{G2}$ and ${\mathrm{MAR}}_{\mathrm{all}}$) obtained higher MAE due to the larger variability of the walking speed of the participants. The MAE distribution of the TO event class in the subsets ${\mathrm{CIR}}_{\mathrm{all}}$ was also slightly higher compared to the MAE of other subsets in the CIRRIS database. The results in the UDS database were stable over all the subsets. Overall, the detection performance of the ESN kept a MAE mostly under 30 ms for the HS and under 60 ms for the TO with the variability of all the walking conditions on each database.

Only the condition of indoor flat running (IR) increased the MAE distribution slightly over 110 ms, as shown in the leftmost boxplot of Figure 9a. The average of the MAE distribution was also significantly above its median. This was because the bias prediction errors of these few timeseries under the IR condition was really different compared to the others. Recentering the error of all the timeseries with a uniform bias equal to the median error of all the timeseries (see Section 2.5) causes a high error for these few that are badly centered according to their respective median. The resulting MAE for these few timeseries were much larger, increasing the mean MEA, but did not affect the rest distributions over all timeseries. Similar cases of higher mean value of the MAE distribution were also observed in the UDS database. In all cases, we see on the right-side axis that the values of the TPV and the PPV were very close from each other, which means that the number of FP and FN was similar. This confirms that the choice of the time threshold of 0.65 τ was adequate.

### 3.3. Performance between Orientations and Positions

This section presents the MAE, TPR, PPV and T1 results for all the event classes. Figure 10, Figure 11 and Figure 12 are boxplots of the MAE grouped individually by event class. Each category on the horizontal axis represents a different choice of axes for the accelerations used as the input between the AP, ML and V orientations (as shown in Figure 2). The first three categories were defined using one orientation of the acceleration ($N_{\mathrm{in}}$ = 2). The next three categories denoted with two axes were defined using two orientations of acceleration ($N_{\mathrm{in}}$ = 3), and the last category was defined with the three orientations ($N_{\mathrm{in}}$ = 4). The selected acceleration axes were the same for both the training part and the testing part.

We observed variable performance of the ESN according to the sensor localisation. The results in Figure 10a were obtained with an input defined by the acceleration recorded on the foot. For the results in Figure 10b,c, the input was defined with the accelerations recorded on the waist and the wrist, respectively, and gave significantly poorer results than the foot positions. The median MAE when the foot acceleration was used as input was between 17 and 54 ms, and the T1 score was mostly over 0.75, compared to approximately 170 and 230 ms and 0.3 when the waist or the wrist accelerations were used as the input. These results show that the ESN GED has much higher performance with foot acceleration signals than upper body accelerations.

We also observed small differences in GED performance according to the choice of axes for the accelerations used as the input. The categories using two or three axes did not seem to offer a significant improvement in performance compared to the categories using a single axis. The best result obtained using only one axis was the AP, and the input categories with several channels which included the AP axis were also more efficient.

### 3.4. Performance According to Various Gait Event Classes

The performance of the ESN varies significantly over the different gait event classes. Observations for each gait event class are listed below: (1) The HS event class obtained the best predictions. The upper whisker of the box plot remained under 30 to 50 ms for the three databases and almost all acceleration axes, and the T1 scores were over 0.92. (2) The HP event class (in the CIRRIS database) also obtained excellent prediction, i.e., with higher whiskers of the MAE values of 50 ms and T1 scores of 0.94. (3) The FF and HO event classes in the CIRRIS database and in the UDS database were predicted with lower accuracy. All acceleration axes for these event classes had an upper whisker of 100 ms (except for the V axis of the HO in the UDS database, which exceeded 200 ms) and a T1 score between 0.6 and 0.8. (4) The TP event class was predicted with variable performance depending on the database. The MAE distributions in the CIRRIS database were quite stable with respect to the different acceleration axes and had an upper whisker of 100 ms and a T1 score of 0.9. In the UDS database, the error results were more variable, with an upper whisker ranging between 70 and 200 ms for various acceleration axes and T1 scores of 0.75. (5) The TO event class prediction performance were also variable according to the database. Results for the MAREA were worst, with MAE distributions up to 120 ms and T1 scores of 0.85. The distributions were clearly better in the CIRRIS database, where the MAE distribution was under 50 ms and the T1 score was over 0.95. The results in the UDS database were almost as good as for CIRRIS with an upper whisker of the distribution of 60 ms and a T1 score of 0.93.

In summary, the event classes at the start of the stance phase, the HS and the HP were better predicted, with a MAE below 50 ms. The event classes in the middle of the stance phase, the FF and the HO, were predicted with less precision, with a MAE of up to 100 ms. The gait events in the end of the stance phase (i.e., TP and TO) were predicted with high variability, the MAE varied from values just below 50 ms to values up to 120 ms.

### 3.5. Algorithms Comparison

We compared the ESN with four algorithms from the literature. We first compared it with our implementation of the TKEO over the three different databases. The TKEO algorithm had similar characteristics as ESN: it required a low number of assumptions and only a few calculation steps were needed to compute the HS and TO event predictions. A second comparison was made with three other algorithms that required complex calculation steps and gave higher levels of precision compared to most of the results seen in the literature.

These three comparative algorithms have been selected because their results, taken from [15], are based on the same data and the same walking conditions as three of the subsets defined in the MAREA gait database. This allows a fair comparison with the ESN. Moreover, these three algorithms have different characteristics that we want to compare:The KH [15] has a good precision. It is one of the best GED algorithms for IMU analysis that provides some of the most precise detection in the literature.The Ru [5] can detect more than two conventional gait events (HS and TO) since it was designed for functional electrical stimulation (FES).The Au [16] is also a fairly simple algorithm that needs only a few filtering and logical decision steps and is based on a few parameters tuned empirically.

#### 3.5.1. Comparison with TKEO

Figure 13 compares the ESN and the TKEO results of the distribution of MAE for the three databases obtain on all the walking conditions (the subsets ${\mathrm{MAR}}_{\mathrm{all}}$, ${\mathrm{CIR}}_{\mathrm{all}}$, ${\mathrm{UDS}}_{\mathrm{all}}$ of Table 1). The input acceleration from the AP axis of the foot, as was proposed in the original TKEO method [10]. The ESN was trained specifically, using training data of one database at a time.

In the MAREA database, the ESN and TKEO algorithms performed in a similar way. For both algorithms, the MAE distribution was under 30 ms for the HS events and mostly under 100 ms for the TO events. A T1 score over 0.92 was obtained for the HS and 0.85 for the TO. On the two other databases, the TKEO algorithm was highly inefficient compared to the ESN. The TKEO had a MAE of up to 500 ms and a T1 score around 0.6 compared to 50 ms and around 0.94 for the ESN. The MAE of the TKEO in the UDS and CIRRIS databases represented an error equivalent to half of the walking cycle.

Figure A3 provides more comparative results of the two algorithms with the other walking conditions subsets presented in Table 1. The ESN algorithm thus showed robust prediction capabilities over all databases compared to TKEO. Figure A3 shows similar comparative results for the two algorithms when specific walking conditions are considered in each of the databases.

#### 3.5.2. Comparison with Kh, Ru and Au

Figure 14 shows results taken from the literature [15] for three other algorithms and their comparison with the performance of the TKEO and ESN algorithms. The same walking conditions of the MAREA database were used (the subsets ${\mathrm{MAR}}_{\mathrm{TW}}$, ${\mathrm{MAR}}_{\mathrm{OW}}$, ${\mathrm{MAR}}_{\mathrm{OWR}}$ of Table 1). The results for the event classes HS and TO are presented with different walking conditions with the $\mathrm{MAE}{}_{i}$ distribution and by the TPR and *SPF* values. The $\mathrm{MAE}{}_{ik}$ distributions are described by their mean and their standard deviation.

In order to ensure a fair comparison, the results of the MAE given for the TKEO and ESN were evaluated in the same manner as other results from the literature [15], using the error of only correctly detected events with a threshold of 40 ms.

The Kh and Ru algorithms were more precise and less variable than the other algorithms. In the Kh algorithm, the detection was almost perfect with a true positive threshold of 40 ms and a TPR and *SPF* always around 0.99. For the Ru algorithm, the detection was also almost perfect in the IW and OW subsets, but a lower accuracy was obtained in the OWR subset, where running conditions were considered. The Au algorithm was oversensitive for TO events, considering a low *SPF* value in the three subsets, meaning that many false positive TO events were detected.

The TKEO and the ESN implementations obtained equivalent results. The MAE distributions of the TKEO and the ESN were mostly higher than the three other algorithms (here on the correctly detected events, with errors under 40 ms only). The maximal difference of the mean MAE was about 10 ms larger in the case of the TO in IW subset. They obtained poorer detection performance than the Kh and Ru algorithms in the IW and OW subsets, but they kept approximately the same results in the OWR subset, contrarily to Kh and Ru, showing that they were less influenced by the walking speed variations. The detection rate of the ESN and the TKEO was always acceptable (0.75 of TPR in the worst case) for all conditions, contrary to the Au and the Ru algorithms that showed oversensitivity for TO even in the subset ${\mathrm{MAR}}_{\mathrm{OWR}}$.

## 4. Discussion

We discuss here five important aspects of the application of the ESN for GED: (A) Training characteristics of the ESN have been found to be critical; (B) The ESN robustness over various aspects of input data; (C) The ESN robustness over different types of gait event predictions; (D) Simplicity of the ESN compared to other state-of-the-art algorithms; (E) The clinical perspectives and some further investigations needed regarding the ESN algorithm.

### 4.1. Training Characteristics to Ensure Good Performance

#### 4.1.1. Minimum Training Duration

A total of 2000 gait cycles was a good standard to obtain the smallest MAE over three different databases containing at least 40 (for the case of the ${\mathrm{CIR}}_{\mathrm{all}}$) different walking conditions and participants, as shown in Figure 7. This means that each participant needs to perform about 50 steps (less than 1 min) of continuous walking in each expected condition to ensure stable performance. If more variability was considered in the training data from other pathologic gaits or with new walking environments, this minimal training duration might need to be increased.

#### 4.1.2. Training for Specific vs. General Applications

Multiple walking conditions, sensor locations and saturation characteristics induce many differences in the amplitude and the signal shapes over the three databases. These differences can significantly increase the prediction error for the ESN. Changing gait data conditions between the training and testing sets was the main issue leading to decreased performance. With specific training data, the ESN precision is maximized for similar testing data. However, we observed in Figure 8 that the mean absolute error can increase by over 10 times if a condition unseen in the training is used later in testing. We also observed, in the last row in the matrices of Figure 8, that the ESN was able to generalize event predictions over all datasets when the training set considered all databases. Generalization, in this application, indicates the possibility to maintain the ESN accuracy for GED even with datasets with high variability and different sources. We can see that high variability in the training data also slightly increases the MAE. In other words, the generalization capability of the ESN is obtained with a small tradeoff in terms of precision. This work showed that both general and specific training data could be advantageous to ensure good GED performance, depending on the application and the training data availability. The general training data need to be sufficiently variable under the multiple conditions that can appear in applications and could ensure the robustness of the ESN under all of these conditions. The specific training data should be sufficiently close to one condition of use (same user, devices, body position and walking conditions) and could ensure higher precision for this specific condition.

### 4.2. Robustness over Various Types of Input Data

#### 4.2.1. Walking Conditions

Figure 9 showed that ESN obtained good predictions over all the walking conditions tested for the HS and TO events. The HS event was predicted with equivalent MAE over all conditions and database. For the TO event, the performance obtained was more variable, as presented in the Section 2.1, and appears to be related to the variability of acceleration profiles during TO events. A peak in the acceleration signal is visible in each gait cycle happening at the TO event, but it takes various forms depending on the database, patient and walking condition. We can see such differences in Figure 4, where the shape of the acceleration signal at the end of the stance phase is different between the UDS, CIRRIS and the MAREA databases. Increasing these variabilities in the shapes seemed to affect the consistency of the prediction performance because, as shown in Figure 9, the performance of the TO event detection clearly degrades with the amount of variability in the walking conditions in the three databases. In the MAREA database, the time series had 11 subjects with 5 walking conditions and 9 subjects with 2 outdoor walking conditions. In the CIRRIS database, only 2 subjects performed 4 simulated walking patterns at 5 continuous and regular speeds on a treadmill. In the UDS database, all 14 subjects walked on 3 different tracks with a self-selected speed level. We can therefore qualitatively see that the variability of walking conditions seems to be smaller in the CIRRIS database, larger for the UDS database and largest for the MAREA database. The performance of the TO predictions (Figure 10a, Figure 11 and Figure 12) followed this level of variability (CIRRIS performs best, followed by UDS, then MAREA).

#### 4.2.2. Sensor Position

The input defined by the acceleration of the foot is more relevant for GED then upper body accelerations, as shown in Figure 10b,c. As discussed in the literature [12], the acceleration of the upper body is more disturbed by parasitic movements uncorrelated with the gait cycle. The prediction of a walking event with a wrist or a waist sensor represents a more complex task. We saw similar observation in the literature [10] that highlighted the difference in prediction performance between algorithms exploiting upper body accelerations and those exploiting lower body accelerations.

#### 4.2.3. Sensor Orientation

The orientation of the accelerometer also influenced the ESN performance. A uniaxial acceleration along the AP axis located on the foot was the best choice to optimize the precision over all types of events and for the three databases. Defining input with additional axes obtained the equivalent error and did not seem to add significant information extractable for the RC. The AP acceleration is usually the primary acceleration considered in other types of GED algorithms [10,11] and can be considered a highly relevant signal for the gait cycle. Nevertheless, the ESN is robust enough to work for GED with multiple axis orientations and combinations because the MAE results remained the order for all the different axes configurations and combinations of inputs tested.

#### 4.2.4. ESN Adaptability Compared to TKEO

Figure 13 shows that the ESN was much more robust than the TKEO algorithm over all the walking conditions included in the three databases. We observed equivalent predictions from the ESN and the TKEO algorithms on the MAREA database, but on the UDS and the CIRRIS databases, only the ESN was able to make reliable predictions.

The lack of robustness of the TKEO algorithm is explained by the way the algorithm distinguishes the HS from the TO events. As explained in detail in the Appendix E, the prediction of the TKEO depends on the assumption that the acceleration signal has a higher peak at the heel strike than at the toe-out. This assumption is true for the MAREA database (the sensor was placed at the ankle), but not in the two other databases (the sensors were placed on the middle top of the foot).

In addition, the TKEO is not a trainable algorithm, a fundamental difference compared to ESN. Physical hypotheses supporting the TKEO algorithms cannot be changed without making major modifications in the algorithm, and would need recoding a hardware system at a low level. Such changes to allow better detection under new conditions require a complex and time-consuming engineering process and cannot realistically be done for each new walking condition considered. In comparison, the ESN training on few data of each database is a quick process that maintains optimal performance in each database without changing anything to the structure of the algorithm. Additionally, the ESN predicts each type of event independently, thus avoiding misdetections caused by errors from other types of events. It is thus easier to adapt the ESN for various types of acceleration signals.

### 4.3. Robustness for Multi-Event Detection

#### 4.3.1. Capability Comparison with Other Algorithms

The ESN also offers the distinct advantage of being trainable for a high number of events, which is not the case for most of the other methods. The Kh algorithm [15] and the TKEO [10] are limited to the detection of the HS and TO only. These event types are the most important in practice, used to identity the swing phase and the stand phase, but other clinical purposes can benefit from the detection of a higher number of event types.

The Ru algorithm [5] was created for real-time functional electrical stimulation (FES) applications, where the distinction of four gait phases is needed. This method uses a rule-based state machine to detect a higher number of events inside the gait cycle. Transition criteria between events that follow the interrelated logic of the gait cycle are used to establish four different events and avoid misdetection between the events. The results of [5] showed that the HS is precisely detected, but mid-stance, pre-swing and swing transition phases, similar to the FF, HO and TO events, are harder to detect, with errors extending, respectively, around 50, 100 and 150 ms of the interquartile range. Results obtained with the ESN were as good as Ru or better, as shown in Figure 10, Figure 11 and Figure 12, where the distribution range of all errors remained between 50 and 120 ms for all the events. This range of error (5–10% of the gait cycle) is less than the error (13% of the gait cycle) tolerated for FES, as proposed in a previous study [43]. This shows the capability of the ESN to detect a high number of events with the required precision for applications such as FES.

Another limitation mentioned in [5] is related to the thresholds and transition delay criteria needed to detect pre-swing events (similar to the HO). These are set empirically with offline processing and could become invalid if variations occurred in the speed of walking. Additional work to establish rules for adaptation of the model could be avoided by the systematic training-based method of the ESN.

#### 4.3.2. Expected Performance According to Event Classes

We demonstrated that the ESN can predict up to six gait event classes (HS, HP, FF, HO, TP and TO) but with different performance for each of them, as discussed in Section 3.3. The maximal value of the MAE over for all walking conditions is between 30 ms for the easiest event to detect (HS) and 120 ms for the hardest events (FF, HO). We looked at the shape of the walking acceleration signals, as illustrated in Figure 4, to interpret this dispersion in the predictive performance. Gait events that happen at the beginning of the stance phase (HS and HP) are much more precisely detected. These events occur when the foot encounters the ground. The resulting impact creates the largest peak in acceleration through the walking cycle. These events are then easily distinguishable by selecting the highest acceleration peak of the gait cycle. Events happening in the middle of the stand phase (FF and HO) were less precisely detected. They appear at the center of the stance phase when the foot is static on the ground and no features in the acceleration can be captured. Their prediction is therefore a more complex challenge for an ESN, as we observed in our results.

We found that TO events, occurring at the end of the gait cycle, obtain variable performance. As discussed in Section 4.2.1, the foot movement is important during this part of the gait cycle, creating a peak in the acceleration, like the HS. However, this peak is more variable in shape and in amplitude depending on the walking condition and database inducing variability in the performance.

Finally, we observed two different level of precision in the result of the TP event for the two databases tested. A lower error was obtained for the CIRRIS database (60 ms MAE) and a higher one was obtained for the UDS database (120 ms MAE). We interpreted that this difference might be caused by the difference of the ground truth defined y for each database. The ground truth of TP events in the CIRRIS database was defined by the second peak of ground reaction force sensed by the treadmill. The ground truth of TP in the UDS database was defined by the peaks of FSR1 measurements positioned under the toe surface, as shown in Figure 2. These two peaks did not occur exactly in the same part of the stance phase, as shown in Figure 4d. In the CIRRIS database, the TP occurred in the middle of the stance phase, when the foot was still immobile on the ground and the acceleration was still zero, such as the FF event class. This event was predicted with similar performance as the FF event class. For the UDS database, the TP occurred later in the stand phase, at the start of the swing movement, when the acceleration signal was non-zero, and its prediction was much better rather like the TO.

According to these interpretations, we can conclude that the events close to prominent features of the accelerations are easier to predict. The events located in the gait cycle where the acceleration signal is low or more variable from one cycle to another are more difficult to detect.

### 4.4. Simplicity Compared to Other Methods

One of the main advantages of the ESN is its simplicity without significant compromise on accuracy. The three other algorithms presented in Figure 14 were based on multi-step filtering and data processing methods. The Kh algorithm from Khandewal [15] uses wavelet transforms to individually detect the nominal cadences, the HS and the TO walking events within separate parts of the time-frequency domain. The search space of event peaks is continuously adjusted according to the nominal cadence, giving robustness and precision for high cadence variability. Small delays are necessary to apply a Gaussian fit over a complete gait cycle to identify peaks without noise perturbations. The Au algorithm [16] exploits a process based on a continuous wavelet transform followed by a locality preserving projection [44] to reduce data dimensionality. The detection of the events is made by a training methodology using a probabilistic classification based on a Gaussian mixture model. The Ru algorithm [5] is based on a multi-step filtering process to locate peaks and a rule-based state machine to maintain the logical order of detection between events. The Ru algorithm is developed by iterative empirical process trials.

For the ESN model, most of the computation complexity is to optimize the hyperparameters (HP) (Appendix A) and formatting the training data (Appendix C) of the ESN. These tasks can be performed before the training or operation phases. The HP is usually set by computing many times both the training and the testing steps of the ESN using one specific set of HP at each time. Optimization methods, such as Bayesian optimization [45], can be used to converge iteratively to an optimal set of HP and can take multiple hours of computation. Once the HP is set, the computation cost to apply the ESN is very low. The training can be done offline and takes less than a second for a 100 node reservoir. The calculation steps of the ESN are even simpler in operation mode. Once the training is done, only the Equations (1)–(3) are used, with a complexity of $O(N^{2}+14N)$ operations for *N* the number of nodes in the reservoir (see Appendix F). An implementation of the ESN in a Feather Huzza ESP32 [46] could complete these calculations at 400 Hz. The memory needed to implement the matrices used in the reservoir, $W_{\mathrm{in}}$, W and $W_{\mathrm{out}}$, is also low. In the case of *N* = 100 nodes, it can be as low as 1 Kbytes. Finally, a low-complexity conventional peak finder algorithm is also needed to identify peaks from the prediction of the ESN prediction. The integration of this algorithm into portable hardware systems is therefore fairly straightforward.

Under normal walking conditions indoors and outdoors, the ESN is less performant regarding the mean and standard deviation of the MAE compared to these 3 other algorithms. However, the ESN seems to be more robust over all conditions since it does not have any oversensitivity problems for the TO detection, as the Au and Ru algorithms do. The ESN also give similar errors even for the conditions with more variability (OWR), contrary to three others that show higher mean results.

### 4.5. Clinical Perspectives and Further Investigations

The ESN algorithm has several practical advantages for its use in clinical applications. The ESN can process GED in real time with low temporal delay, and the number of event classes predicted does not increase this delay. The ESN also has the advantage of adapting to each participant. Gait variability is a complex topic that can be related to multiple independent metrics [47]. Even for healthy walkers, many people have a unique gait pattern [48]. Gait variability and stride regularity can be affected by the age and the walking speed of the subject [49]. Since the training of the ESN is a quick and systematic procedure, it could therefore be useful to individualize the training by calculating the output weights $W_{\mathrm{out}}$ specific to each participant using data obtained during a clinical phase prior to the use of the ESN GED system. This allows to improve the ESN prediction efficiency for participants who have a very specific walking pattern.

This leads to further investigations needed to test such algorithm in more various types of gait. The results presented here were obtained from healthy participants only. GED performances are yet to be evaluated on patients with various conditions affecting their gait. We plan to apply the ESN for GED in a pathological population. As an example, individuals with knee osteoarthritis present heterogeneous and altered gait patterns that could influence the performance of the ESN model [50,51,52]. The comparison of the performance between healthy and pathologic groups of participants will allow further evaluation of the ESN’s capability and robustness.

Lastly, better positioning of the sensors could also improve the ESN’s capability for GED. The positioning of the sensor has been tested for only two positions on the foot and two positions on the upper body, but prediction performance elsewhere on the body remains unknown.

## 5. Conclusions

To conclude, we show in this work that the ESN is an efficient and robust algorithm to detect gait events using the acceleration signal measured by a single-axis IMU system. The ESN can be used for various types of gait events simultaneously and achieve detection with acceptable errors over various walking conditions. The ESN had a MAE at most 10 ms higher compared to four state-of-the-art algorithms on indoor and outdoor walking conditions. Unlike these algorithms, the application of the ESN detector, after training, requires limited computing resources and embedded memory. The ESN is thus easily adaptable for hardware implementations on small microcontrollers to be used in real-time clinical applications, such as FES or other walking monitoring systems. This confirms the applicability of the ESN for GED and even other gait analysis computations that need to be done in real time in wearable (embedded) systems. This also opens the door to the applications of new relevant technology such as the neuromorphic micro electronic mechanical systems (MEMS) accelerometers [53,54] designed to capture one axis acceleration signal and process it using a physical RC micro-mechanical system. Compared to ESN, the neuromorphic MEMS have the advantages of significantly increased autonomy and decreased size in wearable gait analysis devices. Like ESN, the neuromorphic MEMS are further expected to provide more robust, trainable GED capabilities compared to other popular algorithms.

## Figures and Tables

**Figure 1 sensors-22-07180-f001:**
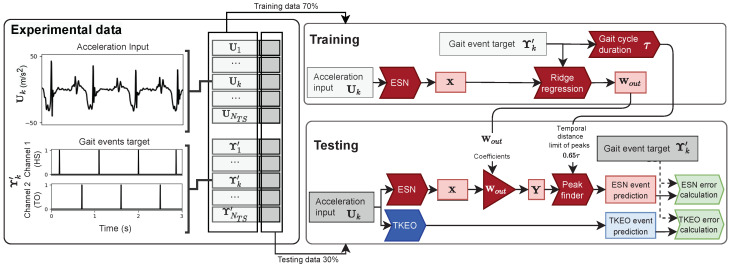
Procedure used to evaluate the performance of the ESN and the TKEO algorithms. Arrow boxes represent calculation steps and square boxes represent data. The experimental data were collected during walking tests as explained in Section 2.1 and preprocessed as explained in Section 2.2. The input timeseries $U_{k}$ is acceleration signals recorded on the participant left foot. The output timeseries $Y_{k}^{\prime }$ is the gait event targets identified with ground force sensors and formatted as a multi-channel binary signal. Each channel indicated the event time indices of one gait event class. The timeseries lists of the inputs $\left\{U_{1},\ldots ,U_{k},\ldots ,U_{N_{\mathrm{TS}}}\right\}$, and the outputs $\left\{Y_{1}^{\prime },\ldots ,Y_{k}^{\prime },\ldots ,Y_{N_{\mathrm{TS}}}^{\prime }\right\}$ represent a subset of multiple input and output timeseries for each participant and walking condition considered in the procedure. The timeseries are separated in two groups, one for the training step of the ESN algorithm and one for the testing step of both ESN and TKEO algorithms. The ESN algorithm steps are represented by the red boxes and explained in Section 2.3. The TKEO algorithm steps are represented by the blue boxes and explained in Section 2.4. The green boxes evaluate the error between predicted and target events and establish performance criteria as presented in Section 2.5.

**Figure 2 sensors-22-07180-f002:**
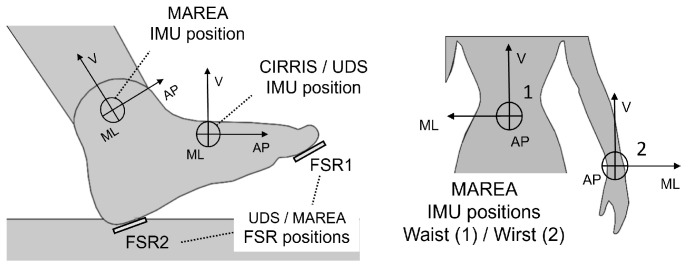
IMU and FSR positioning for the three databases. In CIRRIS tests, the IMU was placed outside the shoes using straps. For MAREA tests, the three IMU were attached with straps at the three positions shown, and the FSR were placed into the shoe soles. In UDS tests, the IMU and the FSR were placed on a sock inside the shoes. The orientations are defined as the antero-posterior (AP), the medio-lateral (ML) and the vertical (V) axis.

**Figure 3 sensors-22-07180-f003:**
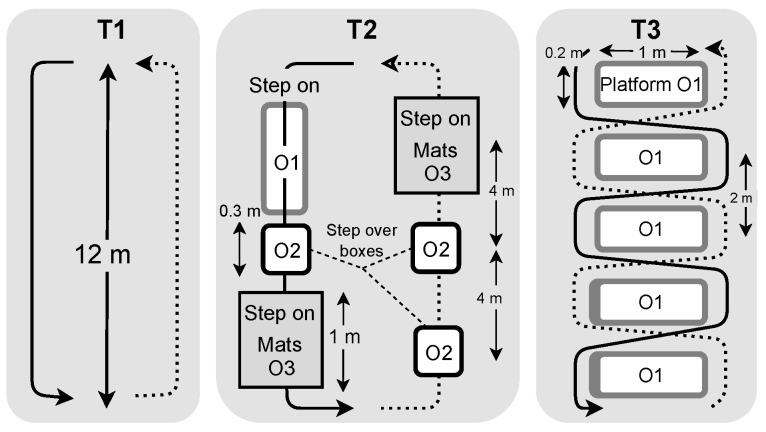
Walking tracks of UDS walking tests. The subjects were walking continuously on the full arrow and coming back following the dotted arrow. T1 was a normal straight walk without obstacles. T2 was a walking circuit over small obstacles, including stepping on one 0.3 m wide by 1 m long by 0.2 m high hard platform (O1), stepping over 0.1 m high by 0.3 m square side platforms (O2) and stepping on 0.5 m wide by 1 m long by 4 cm thick exercise mats (O3). The participant stepped on O1 and O3 by putting their feet on it but had to pass over O2 without touching it. T3 was a walking circuit around 0.3 m wide by 1 m long platforms (O1) aligned in a row with 2 m distance between platforms.

**Figure 4 sensors-22-07180-f004:**
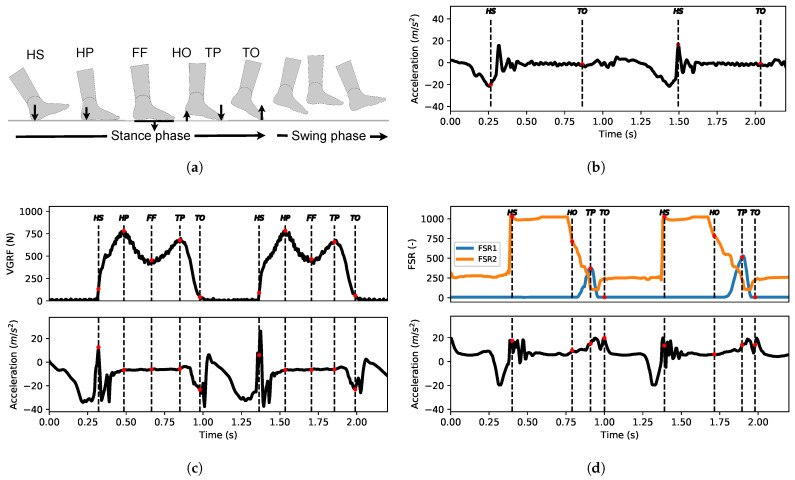
Event target identification process from foot pressure measurements for each three databases. Examples of foot pressure measurements, acceleration measurements, and target gait events identified in two gait cycles for each database are shown in subfigures (**b**–**d**). The heel strike (HS) is the first contact of the foot with the ground. The heel push (HP) is the maximum load response of the foot after the HS. The foot flat (FF) is the minimum load response of the foot in the middle of the stance phase. The heel off (HO) is the end of the heel contact on the ground in the second part of the stance phase. The toe push (TP) is the second maximum load response of the foot in the second part of the support phase. The toe off (TO) is the final contact of the foot on the ground. (**a**) Event identification; (**b**) MAREA database (acceleration of the foot position); (**c**) CIRRIS database; (**d**) UDS database. FSR1 is on the toe and FSR2 is on the heel.

**Figure 5 sensors-22-07180-f005:**
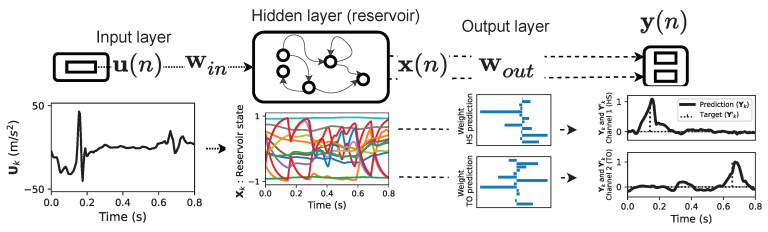
ESN algorithm representation considering the HS and TO gait event classes for the output prediction.

**Figure 6 sensors-22-07180-f006:**
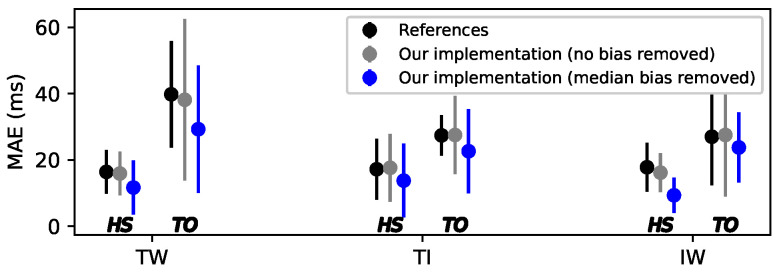
Average (dots) and ±1 standard deviation (error bars) of the MAE for the values reported in reference [10] and our own implementation of the TKEO algorithm, under conditions of treadmill walking (TW), treadmill incline (TI) and indoor walking (IW).

**Figure 7 sensors-22-07180-f007:**
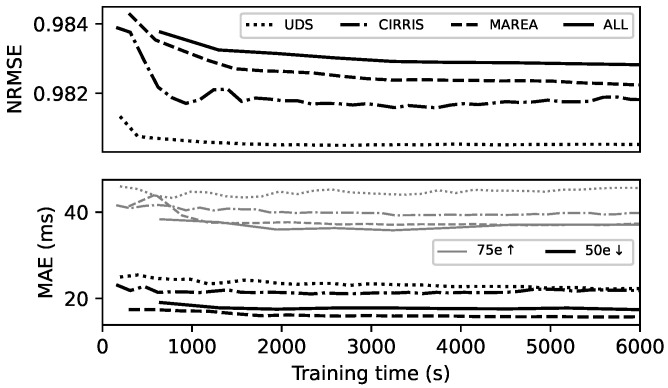
NRMSE and MAE as a function of the length of the training set (with the AP axis acceleration input) considering all events for each database. The errors were evaluated using the ${\mathrm{MAR}}_{\mathrm{all}}$, ${\mathrm{CIR}}_{\mathrm{all}}$ and ${\mathrm{UDS}}_{\mathrm{all}}$ subsets. The three subsets were considered together in the set ALL, with the HS and TO gait event types only. The MAE and NRMSE are different for all databases, mostly because of the harder and higher number of gait event types detected in the CIRRIS and UDS databases than in the MAREA.

**Figure 8 sensors-22-07180-f008:**
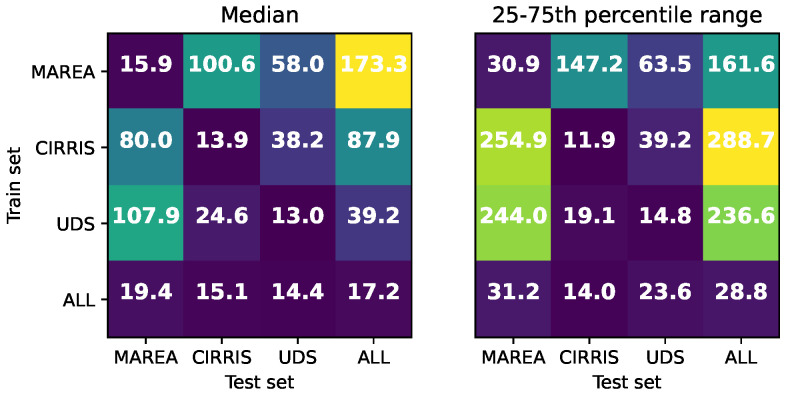
MAE (ms) of the prediction with the ESN on HS and TO events as a function of the training/testing database combination considering all walking condition subsets ${\mathrm{MAR}}_{\mathrm{all}}$, ${\mathrm{CIR}}_{\mathrm{all}}$ and ${\mathrm{UDS}}_{\mathrm{all}}$.

**Figure 9 sensors-22-07180-f009:**
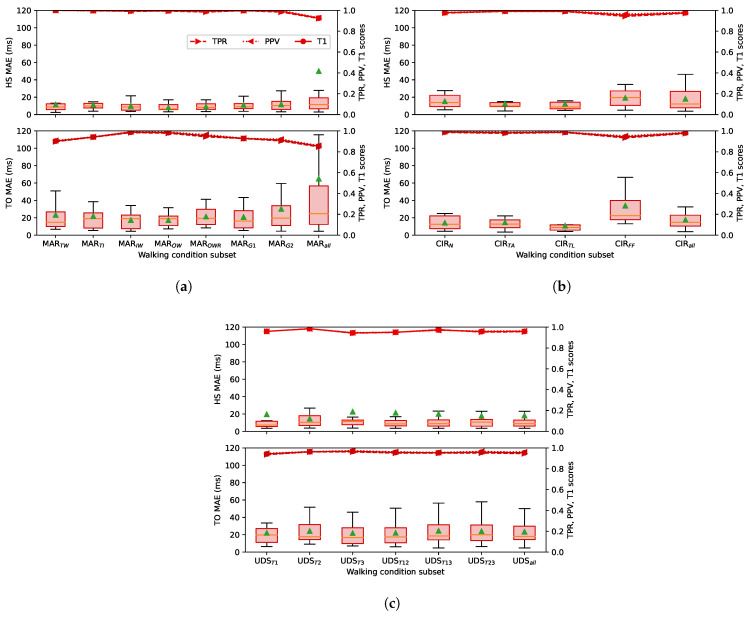
MAE, FPR, PPV and T1 score of the ESN as a function of the subsets of walking condition. (**a**) MAREA database; (**b**) CIRRIS database; (**c**) UDS database. The mean value of MEA is expressed as a green triangle and the median as a central line, while boxes are the 25th and 75th percentiles and the whiskers are the extensions of the last value of $\mathrm{MAE}{}_{ik}$ of the distribution over and under 1.5 times the interquartile range. The values of TPR, PPV and *T*${}_{1}$ are plotted with three lines and are relative to the right-side axis. (Same for all boxplot).

**Figure 10 sensors-22-07180-f010:**
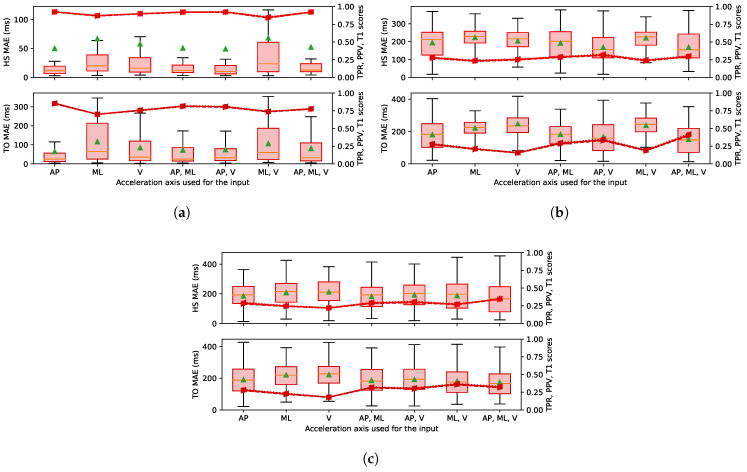
MAREA database results of MAE, TPV, PPV and T1 performance of the ESN for GED on subset ${\mathrm{MAR}}_{\mathrm{all}}$. The boxplots are related to the left axis and represent the MAE distribution over all the timeseries of the testing set. The lines, relative to the right axis, represent the value of the TPV, PPV and T1 score. The horizontal axis categories are relative to different input axis combinaisons. Each vertically aligned subplot represents the results for one event class. The three subfigures represent the result from different IMU positions. (**a**) Foot sensor position; (**b**) wrist sensor position; (**c**) waist sensor position.

**Figure 11 sensors-22-07180-f011:**
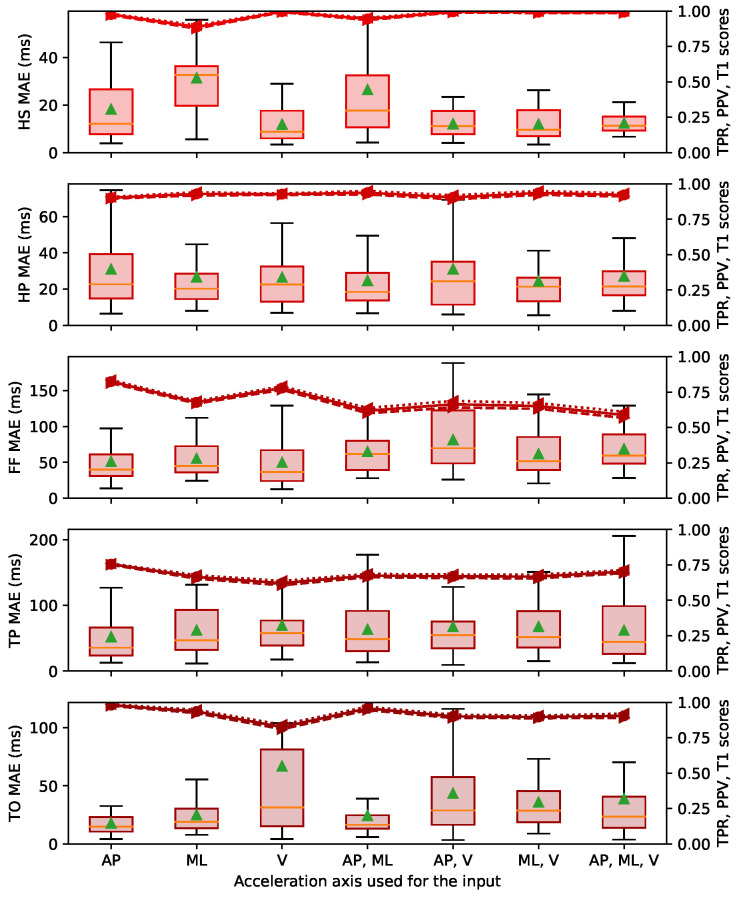
Same as Figure 10 for the CIRRIS database. The ESN for GED on subset ${\mathrm{CIR}}_{\mathrm{all}}$ with foot IMU position.

**Figure 12 sensors-22-07180-f012:**
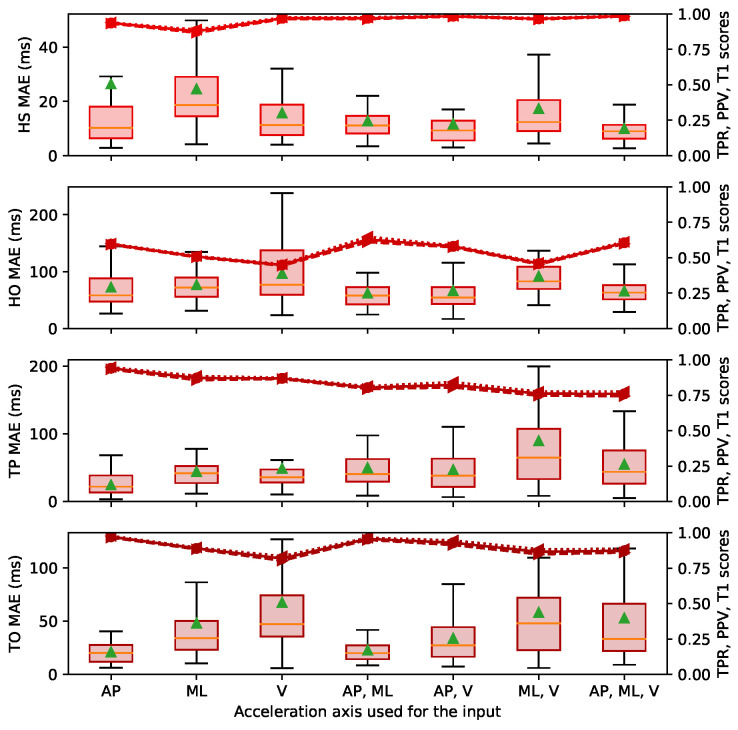
Same as Figure 10 for the UDS database. The ESN for GED on subset ${\mathrm{UDS}}_{\mathrm{all}}$ with foot IMU position.

**Figure 13 sensors-22-07180-f013:**
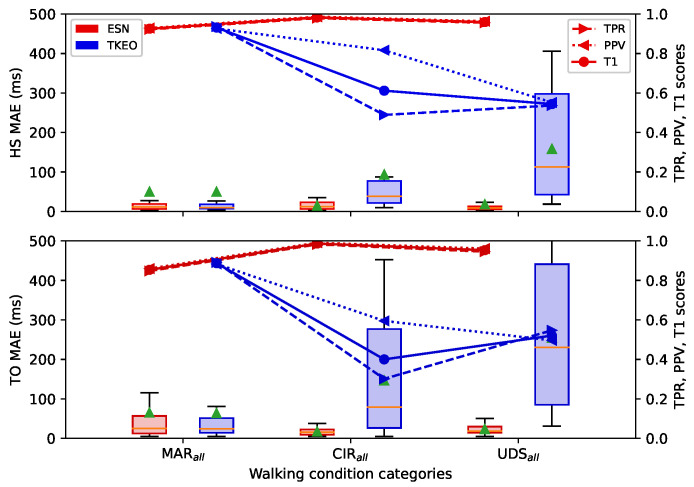
Comparison of MEA, FPR, PPV and T1 score on the three databases for the ESN and the TKEO algorithms.

**Figure 14 sensors-22-07180-f014:**
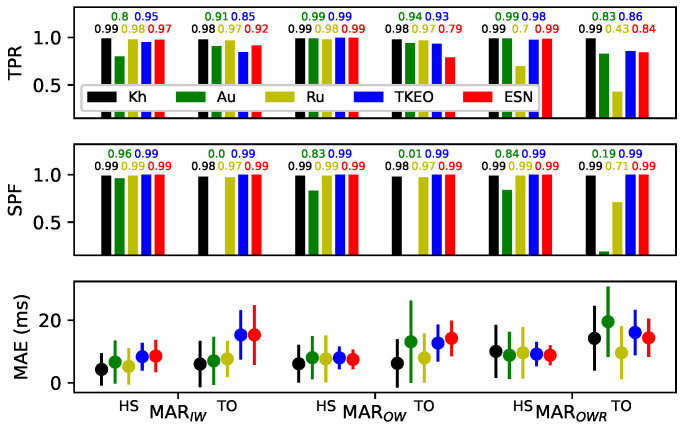
Comparison of the TKEO and ESN algorithms to other algorithms in terms of the TPR, the *SPF* and the mean (central point) and ±1 standard deviation (error bar) of the MAE. The subsets ${\mathrm{MAR}}_{\mathrm{IW}}$, ${\mathrm{MAR}}_{\mathrm{OW}}$ and ${\mathrm{MAR}}_{\mathrm{OWR}}$ of the MAREA database were used, as were the results of the Kh, Au and Ru algorithms from [15]. The performance criteria used here for TKEO and ESN were adjusted to match the criteria used for Kh, Au and Ru. A threshold of 40 ms was used to consider the events as TP. The MAE was evaluated with the error from the TP events only. The number of points in each timeseries was considered for a sampling rate of 128 Hz.

**Table 1 sensors-22-07180-t001:** Database characteristics.

Database	Walking Condition		# Participants	Duration (min)	Speed (m/s)	Type and Number of Timeseries Selected in Each Subset							
MAREA						${\mathrm{MAR}}_{\mathrm{TW}}$	${\mathrm{MAR}}_{\mathrm{TI}}$	${\mathrm{MAR}}_{\mathrm{IW}}$	${\mathrm{MAR}}_{\mathrm{OW}}$	${\mathrm{MAR}}_{\mathrm{OWR}}$	${\mathrm{MAR}}_{G1}$	${\mathrm{MAR}}_{G2}$	${\mathrm{MAR}}_{\mathrm{all}}$
	Treadmill walking	TW	11	≈5	1.1 to 2.2	11					11	11	11
	Treadmill run	TR		≈5							11	11	11
	Treadmill incline walk	TI	11	12	SS *		11				11	11	11
	Indoor flat walk	IW		3	SS			11			11	11	11
	Indoor flat run	IR		3	SS								11
	Outdoor walk	OW	9	3	SS				9	9		9	9
	Outdoor run	OR		3	SS					9		9	9
	Total ($N_{\mathrm{TS}}$ **)					11	11	11	9	18	33	62	73
CIRRIS						${\mathrm{CIR}}_{N}$	${\mathrm{CIR}}_{\mathrm{TA}}$	${\mathrm{CIR}}_{\mathrm{TL}}$	${\mathrm{CIR}}_{\mathrm{FF}}$	${\mathrm{CIR}}_{\mathrm{all}}$			
	Normal walk	N	2	2		10				10			
	Right toe with external angle	TA		for each	0.8, 1, 1.2,		10			10			
	Right trunk leaning	TL		speed	1.4, 1.6			10		10			
	Right foot flat	FF							10	10			
	Total ($N_{\mathrm{TS}}$ **)					10	10	10	10	40			
UDS						${\mathrm{UDS}}_{T1}$	${\mathrm{UDS}}_{T2}$	${\mathrm{UDS}}_{T3}$	${\mathrm{UDS}}_{T12}$	${\mathrm{UDS}}_{T23}$	${\mathrm{UDS}}_{T13}$	${\mathrm{UDS}}_{\mathrm{all}}$	
	Normal walk	T1	15	8	SS	15			15		15	15	
	walk with obstacles	T2			SS		15		15	15		15	
	walk around obstacles	T3			SS			14		14	14	14	
	Total ($N_{\mathrm{TS}}$ **)					15	15	14	30	29	29	44	

* SS: self-selected speed; ** N_TS_: the number of timeseries in each subset is equal to the cumulative number of participants selected in each walking condition. This gives an indication of the amount of data variability in each subset. The larger N_TS_ is, the greater the variability of the gait data is (more patients and more different gait conditions). In the last 8 columns, colored box indicate the walking conditions considered in each subsets. Non-colored box indicate the walking conditions not considered, and the numbers indicate how many records were considered for each walking conditions.

**Table 2 sensors-22-07180-t002:** Echo state network hyperparameters.

Variable Name	Symbol	Value
Number of nodes	N	100
Leaking rate	α	0.1053
Hidden layer spectral radius	ρ	0.7471
Hidden layer sparsity	P	0.21
Input layer scaling	$S_{i}$	2.300
Input layer sparsity	$P_{i}$	0.41
Input layer bias scaling	$S_{b}$	2.911
Regularization parameter	γ	1 × 10${}^{-6}$

**Table 3 sensors-22-07180-t003:** Classification of the target events (TE) and the prediction events (PE) for FN, TP and FP.

1. False Negative (FN) Target event not detected	2. True Positive (TP) Target event correctly detected	3. False Positive (FP) Prediction event incorrect
TE with a time distance higher than 65 ms from the closest PE.	TE with a time distance equal or lower than 65 ms from the closest PE.	PE with a time distance with the closest TE higher than 65 ms or higher than any time distance of other PE and the TE.
$N_{i}^{FN}$ = # { $|E_{ikl}|>$65 ms $|1\leq k\leq N_{\mathrm{TS}},\;1\leq l\leq N_{TE}\left(i,k\right)$ }	$N_{i}^{TP}$ = # { $|E_{ikl}|<$65 ms $|1\leq k\leq N_{\mathrm{TS}},\;1\leq l\leq N_{TE}\left(i,k\right)$ }	$N_{i}^{FP}$ = # {$N_{PE}\left(i,k\right)\;|\;1\leq k\leq N_{\mathrm{TS}}$ } − $N_{i}^{TP}$
**4. True Negative (TN)**		
All remaining points		
The number of remaining time points in the timeseries unselected as a TP, FP or FN event		
$N_{i}^{TN}=\#\left\{N_{T}\left(k\right)|1\leq k\leq N_{\mathrm{TS}}\right\}-\left(N_{i}^{TP}+N_{i}^{FP}+N_{i}^{FN}\right)$		
$N_{T}\left(k\right)$ is the total number of time points in each timeseries		

## Data Availability

MAREA database was obtained from Siddhartha Khandelwal and are available at https://wiki.hh.se/caisr/index.php/Gait_database with the permission of Khandelwal. CIRRIS and UDS databases are available on request from the corresponding author.

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
