# Peer review of "Detecting Gait Events from Accelerations Using Reservoir Computing"

_sensors, 2022, doi:10.3390/s22197180_

Round 1
Reviewer 1 Report
This paper presents a numerical implementation of a Reservoir computing (RC) algorithm called the Echo state Network (ESN), that is based on simple computational steps that are easy to implement on portable hardware systems for real-time detection.
The paper is well written. The experiments are pervasive.
Author Response
The authors thank the reviewer for the detailed comments and for taking the time to carefully review the paper.
No major comments have been specified by your review, but modifications have been applied to the paper according to the comments of the two other authors. The modifications are highlighted in blue in one of the PDF resubmitted filed.

Reviewer 2 Report
This is very interesting article that can be accepted for publication, with only some minor comments, as follows:
- The main findings of the study could be included in the abstract.
- Avoid using this style of citations [1-14].
- Clarify the cost and complexity of your method.
- refresh your references with some recent works, from 2022,such as:
Utilizing Spatio Temporal Gait Pattern and Quadratic SVM for Gait Recognition;
Multi-ResAtt: Multilevel Residual Network with Attention for Human Activity Recognition Using Wearable Sensors;
Advances in Vision-Based Gait Recognition: From Handcrafted to Deep Learning;
VGG16-MLP: Gait Recognition with Fine-Tuned VGG-16 and Multilayer Perceptron;
- A proofreading is needed.
Author Response
The authors thank the reviewer for the detailed comments and for taking the time to carefully review the paper. Modifications have been applied to the paper according to the comments. The modifications are highlighted in blue in one of the PDF resubmitted filed. The following explanations are directly related to your comments:
----------------------------------------------------------------------------------------------------------------------------------------------------------------------------------------------------------------------------------------------------------------
- The main findings of the study could be included in the abstract.
Response : The sentence "Our main finding is that despite its low complexity, ESN is robust for GED, with performance comparable to other state-of-the-art algorithms." was added to the abstract.
Change location: abstract, p.1, lines 12-13
--------------------------------------------------------------------------------------------------------------------------------
- Avoid using this style of citations [1-14].
Response : The bibliography style has been adjusted to the American Chemical Society style proposed by the MDPI journal guide (Reference List and Citations Style Guide for MDPI Journals - https://www.mdpi.com/authors/references)
Change location: bibliography style has been changed for the entire document
--------------------------------------------------------------------------------------------------------------------------------
- Clarify the cost and complexity of your method.
Response : Again, thank you for this relevant comment. We have added additional explanations with quantitative details on the complexity of the methods. This was done by estimating the complexity (in number of operations) and the memory usage (in KBytes) of the ESN method. This information was added in the discussion, under the section '' Simplicity Compared to other Methods ''. Additional details of the complexity estimation were placed in appendix 'F ' Complexity of the ESN'‘.
Change location: discussion, p.23. lines 760 and appendix F p. 31
--------------------------------------------------------------------------------------------------------------------------------
- refresh your references with some recent works, from 2022,such as:
- Utilizing Spate-Temporal Gait Pattern and Quadratic SVM for Gait Recognition;
- Multi-ResAtt: Multilevel Residual Network with Attention for Human Activity Recognition Using Wearable Sensors;
- Advances in Vision-Based Gait Recognition: From Handcrafted to Deep Learning;
- VGG16-MLP: Gait Recognition with Fine-Tuned VGG-16 and Multilayer Perceptron;
Response : As recommended, we updated references and found that these references were interesting to mention in the introduction. Indeed, they show that DL approaches in biomechanics are mostly explored for the popular task of human activity recognition (HAR) to increase applications in the fields of the human machine interface and biometrics. In particular (Masood, 2022.) shows that GED can be used for gait cycle normalization in order to increase performance of HAR algorithms.
Change location: introduction, p.2, lines 70-77
--------------------------------------------------------------------------------------------------------------------------------
- A proofreading is needed.
Response : The article has been revised again in detail.

Reviewer 3 Report
This is a very informative study that showed the novel method for gait event detection (GED) using an inertial measurement unit to apply a class of artificial neural network algorithms; the Echo state Network (ESN). The results indicated that the ESN approach had the simplicity to accomplish GED and provided as precise as the other method. That original method for GED can process in real-time, and thus, the method could be very useful clinically.
However, I have some general questions/comments and minor comments:
General comments
In the Introduction, I think you should describe more why the new method for GED should be studied. I understand that there is no study to apply ESN for GED. However, many methods for GED had been reported already, especially those that could be employed automatically, as precise as acceptable clinically, as you stated in the paper. What were the problems with the methods reported so far, and what are the benefits of demonstrating the usefulness of the methods using the ESN?
In addition, I could not understand the need to identify the other gait events such as HP or HO than HS and TO, although you stated that the novelty of this study was to detect multiple events. Please explain in more detail.
Minor comments
Overall, there were several sentences with missing spaces, containing unnecessary words, or which did not begin with a capital letter. Please recheck.
Methods
p.4, line 148, “right toe out (TO). flat strike,,,” Was the period typo? The abbreviation “TO” was the same as “toe-off”. Please change to avoid confusion.
p.10, I consider that you should describe why you applied the four other methods to compare to the ESN among a lot of reported methods.
Discussion
p.20, line 566, Was the word “to” superfluous?
p.21, In the 4.2.4., you stated that the ESN was more robust than the TKEO, and discussed the lack of robustness is explained by the way of the algorithm. Nevertheless, did the TKEO train the algorithm to integrate all the datasets; MAREA, CIRRIS, and UDS? If it was trained from MAREA only, wouldn't it be natural that the robustness would be reduced? The same situation was shown with the ESN, in Figure 8.
Conclusion
p.24, line 761, The abbreviation “MEMS” first appeared in this article. Please list the words before they are abbreviated.
Author Response
Reviewer 3
The authors thank the reviewer for the detailed comments and for taking the time to carefully review the paper. Modifications have been applied to the paper according to the comments. The modifications are highlighted in blue in one of the PDF resubmitted filed. The following explanations are directly related to your comments:
----------------------------------------------------------------------------------------------------------------------------------------------------------------------------------------------------------------------------------------------------------------
- General comments: In the Introduction, I think you should describe more why the new method for GED should be studied. I understand that there is no study to apply ESN for GED. However, many methods for GED had been reported already, especially those that could be employed automatically, as precise as acceptable clinically, as you stated in the paper. What were the problems with the methods reported so far, and what are the benefits of demonstrating the usefulness of the methods using the ESN?
Response : Thank you for pointing out this important aspect of the ESN. We have added more detail in the introduction to clarify this central question. We are aware that GED algorithms were widely explored in the past, and it is quite hard to make better algorithms in performance and in robustness than what already exists in the literature. However, our main goal in this article is to address the challenge related to many recent wearable devices design for highly integrated and autonomous applications. This challenge is the capability of being robust under uncontrolled variability of walking conditions and being designed with low complexity for easy implementation in hardware systems. We think that these two elements are the main advantages of the ESN. A paragraph in the introduction has been had to clarify this and explain how ESN is a good alternative for that compared to other solutions.
Change location: Introduction, p.2, lines 50-60
--------------------------------------------------------------------------------------------------------------------------------
- General comments: In addition, I could not understand the need to identify the other gait events such as HP or HO than HS and TO, although you stated that the novelty of this study was to detect multiple events. Please explain in more detail.
Response : This is also an interesting aspect that we further discuss in the introduction to inform readers of a few applications where a higher number of gait events are needed. We have added a short paragraph in the introduction.
Change Location: Introduction, p.2, lines 64-69
--------------------------------------------------------------------------------------------------------------------------------
- Methods: p.4, line 148, “right toe out (TO). flat strike,,,” Was the period typo? The abbreviation “TO” was the same as “toe-off”. Please change to avoid confusion.
Response : Yes, this was confusing. we have changed the abbreviations for toe out as ''TO'' and for toe with external angle as ''TA''.
Change location: in section 2.1.3, table 1, figure 9b and figure A.3 b) and c) )
--------------------------------------------------------------------------------------------------------------------------------
- Methods: p.10, I consider that you should describe why you applied the four other methods to compare to the ESN among a lot of reported methods.
Response : This information was added to justify the choice of the selected comparative algorithms at the beginning of the result section of comparison method. The main reason is that they have been used with the same data and the same conditions (MAREA gait database). Some advantages presented by each of these algorithms are also specified to inform the reader of differences between the comparative algorithms selected.
Change location: Results, p. 17, lines 531 - 540
--------------------------------------------------------------------------------------------------------------------------------
- Discussion: p.20, line 566, Was the word “to” superfluous?
Response : Thank you for pointing out this mistake, the sentence is now corrected.
Change location: Discussion, p.20, lines 602-603
--------------------------------------------------------------------------------------------------------------------------------
- Discussion: p.21, In the 4.2.4., you stated that the ESN was more robust than the TKEO and discussed the lack of robustness is explained by the way of the algorithm. Nevertheless, did the TKEO train the algorithm to integrate all the datasets; MAREA, CIRRIS, and UDS? If it was trained from MAREA only, wouldn't it be natural that the robustness would be reduced? The same situation was shown with the ESN, in Figure 8.
Response : Thank you for this question, we appreciate the time you took to analyze such subtle details of the methods. It is true that for both algorithms, attempting to apply a new walking condition unseen before leads to poor detection accuracy. The main clarification that was added is that TKEO is a non-trainable algorithm, contrary to the ESN. To explain this, we changed part of the discussion to explain the fundamental difference between a trainable algorithm like ESN and a non-trainable algorithm like TKEO. We explain that a change in the process of the TKEO would probably lead to a redesign of the algorithm. In comparison, changes to only one parameter matrix (Wout) allows the ESN to work in all conditions.
These clarifications have been added in the discussion in the section '' ESN Adaptability Compared to TKEO ''.
Change location: Discussion, p.22, lines 671-680
--------------------------------------------------------------------------------------------------------------------------------
- Conclusion: p.24, line 761, the abbreviation “MEMS” first appeared in this article. Please list the words before they are abbreviated.
Response : The expended acronym was added to the conclusion to better inform the reader of what kind of technology we are talking about.
Change location: Conclusion, p.25, lines 817
---------------------------------------------------------------------------------------------------------------------------------------------------------------------------------------------------------------------------------------------------------------

Round 2
Reviewer 3 Report
The authors responded appropriately to all of my comments.
Thank you.